# Silicon Microring Resonator Biosensor for Detection of Nucleocapsid Protein of SARS-CoV-2

**DOI:** 10.3390/s24103250

**Published:** 2024-05-20

**Authors:** Yusuke Uchida, Taro Arakawa, Akio Higo, Yuhei Ishizaka

**Affiliations:** 1Graduate School of Engineering, Yokohama National University, 79-5 Tokiwada, Hodogaya-ku, Yokohama 240-8501, Japan; 2System Design Lab, School of Engineering, The University of Tokyo, 2-11-16, Yayoi, Bunkyo-ku, Tokyo 113-0032, Japan; higo@if.t.u-tokyo.ac.jp; 3Department of Science and Engineering, Kanto Gakuin University, 1-50-1 Mutsuura-higashi, Kanazawa-ku, Yokohama 236-8501, Japan

**Keywords:** silicon microring resonator, nucleocapsid protein, optical biosensing, InG antibody, Si-tagged protein

## Abstract

A high-sensitivity silicon microring (Si MRR) optical biosensor for detecting the nucleocapsid protein of SARS-CoV-2 is proposed and demonstrated. In the proposed biosensor, the surface of a Si MRR waveguide is modified with antibodies, and the target protein is detected by measuring a resonant wavelength shift of the MRR caused by the selective adsorption of the protein to the surface of the waveguide. A Si MRR is fabricated on a silicon-on-insulator substrate using a CMOS-compatible fabrication process. The quality factor of the MRR is approximately 20,000. The resonant wavelength shift of the MRR and the detection limit for the environmental refractive index change are evaluated to be 89 nm/refractive index unit (RIU) and 10^−4^ RIU, respectively. The sensing characteristics are examined using a polydimethylsiloxane flow channel after the surface of the Si MRR waveguide is modified with the IgG antibodies through the Si-tagged protein. First, the selective detection of the protein by the MRR sensor is experimentally demonstrated by the detection of bovine serum albumin and human serum albumin. Next, various concentrations of nucleocapsid protein solutions are measured by the MRR, in which the waveguide surface is modified with the IgG antibodies through the Si-tagged protein. Although the experimental results are very preliminary, they show that the proposed sensor has a potential nucleocapsid sensitivity in the order of 10 pg/mL, which is comparable to the sensitivity of current antigen tests. The detection time is less than 10 min, which is much shorter than those of other antigen tests.

## 1. Introduction

The worldwide COVID-19 pandemic has had devastating social and economic effects in almost all countries in the world [1]. The pandemic is caused by the human-to-human respiratory transmission of SARS-CoV-2 [2,3,4]. Currently, the number of infected people is decreasing owing to immunity gains through vaccination and infection, and the pandemic has ended. However, preparation is needed to cope with an increase in the number of infected people due to the appearance of new variants and derivatives. [5]. To this end, early diagnosis and simple, accurate diagnostic methods are required to prevent the spread of the virus through human contact [6,7,8,9].

The current COVID-19 diagnostic methods include polymerase chain reaction (PCR), antigen, and serological antibody tests. PCR is a technique that tests for the presence of viral genes in nasal or nasopharyngeal swab or saliva samples and has high sensitivity and accuracy. On the other hand, it is relatively expensive and involves lengthy processes of viral dissolution, RNA extraction, reverse transcription, and amplification during diagnosis, and sample contamination is likely to occur [6]. It also has the disadvantage of requiring specialized testing equipment, which is not readily available in remote or resource-poor environments [7,10]. The antigen test is a method of detecting spike or nucleocapsid proteins that are unique to the virus in collected samples of saliva and others. Compared with the PCR test, it is cheaper and provides results in a shorter time. However, it has lower sensitivity, accuracy, and reliability than PCR [6]. In the antibody tests, the level of biomarkers levels, such as immunoglobulin M (IgM) and immunoglobulin G (IgG) are measured [5,11]. Although this testing method is very rapid and useful as a point-of-care test, it is not suitable for early diagnosis because serum antibodies are not generated until two or three weeks after the first infection [6]. PCR and antigen tests determine whether the patient is infected at the time of testing, whereas the antibody test determines whether the patient has been infected before the time of testing. Therefore, there remains a requirement for a low-cost, rapid, and sensitive SARS-CoV-2 test method with performance that complements the weaknesses of the PCR and antigen tests. In particular, optical biosensors are promising because they are safe, easy to use, and cost-effective [12]. Various types of optical biosensors have been reported, including sensors based on a ring resonator, a Mach–Zehnder interferometer, surface plasmon resonance (SPR), localized SPR (LSPR), surface plasmon polariton (SPP), optical fiber sensors, and bimodal waveguide (BiMW) sensors, many of which have been developed for COVID-19 [11,13,14,15,16,17,18,19,20]. Among them, we focused on silicon microring resonators (Si MRRs), which are compact and low cost, and can be mass produced [21,22]. As for the features of a Si MRR, deep dips appear at a resonant wavelength, which shifts in response to a change in the refractive index of the surrounding environment [23,24]. Target detection is also possible from this wavelength shift and the estimation of virus amount from the amount of shift. There are only a few reports of sensors that meet the minimum sensitivity of the PCR testing of 1 to 2 digits pg/mL [25], and many of them require labeling. However, by using a Si MRR, it is expected that a small biosensor with real-time, one-step and high-sensitivity detection can be realized without labeling [19,26].

Although various types of high-sensitivity Si MRR biosensors have been proposed and developed thus far [27,28,29], no SARS-CoV-2 virus or nucleocapsid protein sensor has been realized. This is because it is difficult for the viruses and nucleocapsid proteins to be adsorbed onto the MRR waveguide surface, making the highly sensitive and selective detection of coronaviruses challenging.

In this study, we focus on the modification of the Si waveguide surface with the IgG antibodies through Si-tag protein A. We have developed a high-sensitivity Si MRR biosensor for nucleocapsid of SARS-CoV-2 [30]. The nucleocapsid is one of the proteins of coronaviruses and is used as a target for antigen tests. The target protein is attached to the sensor surface through the antigen–antibody reaction, which is a specific adsorption reaction between the antigen and the antibody. We fabricated a Si MRR sensor with a polydimethylsiloxane (PDMS) flow channel and evaluated its performance in detecting the nucleocapsid protein. The MRR sensor with the flow channel has the advantage that sample concentration changes due to evaporation during detection do not occur.

## 2. Device Structure and Operating Principle

Figure 1 shows schematic images of the proposed Si MRR sensor. A Si busline waveguide is coupled to the MRR through a directional coupler. The optical power coupling efficiency of the directional coupler *K* is designed to be 0.05. The round-trip length of the MRR is set to 90 μm. The Si waveguides are modified with the IgG antibodies through Si-tagged protein A, as discussed in Section 4.1. The thicknesses of the SiO_2_ upper and lower cladding layers are 3 μm and 700 nm, respectively. The upper cladding SiO_2_ layer directly above the MRR and the neighboring part of the busline waveguide are removed, and a sensing window is formed. The cladding layer is left on the rest of the busline waveguide to reduce the light propagation loss of the busline waveguide, which can be caused by the IgG antibodies and nucleocapsid protein adsorbed onto the surface of the waveguide. The thickness and width of the Si waveguide are 220 and 500 nm, respectively.

Optical biosensors mainly target proteins or ions through the change in the effective refractive index of the waveguide caused by the adsorbed proteins or ions onto the sensor part. Generally, to selectively detect a target, a substance that shows specific adsorption to the target is attached to the sensor surface in advance. When detecting viruses, the antigen–antibody reaction is used. The antigen–antibody reaction is also called the immune reaction, which is the basis of bioinstrumentation and it is used in a wide range of fields, including the detection of viruses, cancer cells, genetic analysis, and environmentally toxic substances. Optical biosensors have the advantage of not requiring the labeling of targets such as proteins or ions.

The MRR sensor consists of a silicon ring or race-track-shaped waveguide and a busline waveguide, which are coupled to each other through a directional coupler. A lightwave guided through the bus line is coupled to the ring waveguide with the optical power coupling efficiency *K*. Figure 1 shows the schematic image of the waveguide of the biosensor developed in this study. The Si core width and height are set to 500 and 220 nm, respectively.

The core of the waveguide is silicon, and the under-cladding is SiO_2_, with refractive indices of 3.455 and 1.445, respectively. In the MRR transmission spectrum, there is a sharp decrease in output only near its resonant wavelength, which can be observed as a resonance dip. The dip shifts in response to the change in the refractive index of the surrounding environment. The resonant wavelength is given by
(1)$$\lambda_{0}=\frac{L\cdot n_{eff}}{N},$$
where *n_eff_* is the effective index, *L* is the round-trip length of an MRR, and *N* is the resonance order. Antibodies are modified on the surface of the Si MRR waveguide, as shown in Figure 1b. The target virus is selectively attached to the antibody on the waveguide surface through the antigen–antibody reaction. The effective refractive index changes, and the resonant wavelength of the MRR changes accordingly to detect the presence or absence of the target.

## 3. Fabrication

### 3.1. Fabrication of Si MRR

The Si MRR was fabricated on a silicon-on-insulator (SOI) substrate by electron beam (EB) lithography and dry etching. The detailed fabrication process is as follows: Process I, Si waveguide formation; Process II, the Deposition of the SiO_2_ upper-cladding layer and the formation of a sensing window by lift-off, and Process III, dicing.

In Process I, the SOI substrate was cleaned with organic solvents. The stripping solution was heated to 80 °C on a hot plate, and the substrate was soaked in it for 10 min to remove dust and particles on the substrate. Then, the substrate was rinsed with acetone, ethanol, and pure water in this order, and the surface was dried with nitrogen. After the cleaning, a charge-dissipating agent (Aqua Save) and a negative electron beam resist (HSQ) were spin-coated on the substrate. Next, the waveguide of the electron beam resist was patterned using an electron beam lithography system (Advantest F7000S-VD02). The advantages of electron beam lithography are that it has a higher resolution than optical lithography and that it can directly write on the substrate, thus eliminating the need to create masks for each design. After the charge-dissipating coating was removed by rinsing with pure water, the substrate was soaked in the developing solution tetramethyl ammonium hydroxide (TMAH) for 8 min, which was followed by rinsing with pure water. Next, reactive-ion etching was performed to form the Si waveguides using SF_6_ and CHF_3_ gases with ULVAC CE-300I. Although Si is normally isotropically etched with SF_6_, CHF_3_ allows the polymer material to remain on the etching surface owing to the action of the plasma. At the bottom of the etched surface, etching by SF_6_ further proceeds. However, the polymer material acts as a protective film on the etched sidewall, and the etching does not proceed, which leads to an anisotropic etching of Si. In Process II, the adhesion-enhancing material OAP (HMDS) and a photoresist (ZPN1150) were first spin-coated on the substrate, which was followed by the formation of a sensing window pattern formed using a laser lithography system (Heidelberg DWL66+). In an MRR biosensor, the target molecules should be attached to the surface of the MRR for detection. Therefore, a sensing window is designed so that the cladding becomes air only at the top of the MRR. A 700 nm thick SiO_2_ upper cladding layer was deposited on the chip by magnetron sputtering (SIH-450, ULVAC, Inc., Kanagawa, Japan). The SiO_2_ film was removed from the top of the window pattern formed by laser lithography, and a 100 × 200 μm^2^ detection window was formed only on the top of the MRR. The substrate was immersed in the stripping solution at 80 °C for 30 min to enhance the penetration of the stripping solution into the photoresist. Then, the substrate was ultrasonically cleaned with acetone for 10 min, which was followed by rinsing in ethanol and then in a pure water. In Process III, the substrate was cleaved into three chips using a stealth dicer (DFL7340, DISCO Corporation, Tokyo, Japan), and the flat facets were formed at the edge of the substrate for optical fiber coupling. In the stealth dicer, the reforming layer is formed by focusing the laser inside the substrate along the cutting line. Small external stresses can easily cleave the substrate with the reformed layer, forming smooth cleaving facets. The laser beams were scanned along the cleavage line to form the cleavage line. Then, a long, thin rod-like object was placed under the substrate, and the substrate was separated into chips by physical cleaving.

Figure 2a,b show an optical microscopy image and a scanning electron microscopy (SEM) image of the fabricated MRR, respectively. The round-trip length of the MRR *L* is 90 μm, and the designed power coupling efficiency *K* is 0.05.

Optical measurements of the fabricated Si MRR sensors were then performed to evaluate them using Santec sweep test system based on a tunable laser (TSL-570, Santec Holdings Corporation, Aichi, Japan) and an optical power monitor (MPM-210H, Santec Holdings Corporation, Aichi, Japan) with a wavelength resolution of 1 pm.

Figure 3 shows the output spectrum of the Si MRR. The dips due to the resonance in the MRR are clearly observed. The free spectral range (FSR) is 6.3 nm, and the evaluated Q factor is approximately 20,000. The power coupling efficiency of the directional coupler *K* is estimated to be 0.035. The resonant wavelength shift of the MRR per refractive index unit (RIU) and the detection limit for the environmental refractive index change are evaluated to be 89 nm/RIU and 10^−4^ RIU, respectively, considering that the wavelength resolution of the measurement system is 1 pm.

### 3.2. Fabrication of Flow Channel

A flow channel is used to detect the target protein. The target protein is dissolved in a solution and poured onto the upper part of the MRR. The use of a flow channel has the advantages of preventing sample concentration changes due to evaporation and allowing measurements without shifting the optical fiber axis when a solution flows [21]. The flow channel was fabricated using PDMS. After ashing and heating at 80 °C, the channel was attached to the MRR chip, and they were successfully attached to each other without peeling off. The flow channel dimensions are 0.4 × 0.8 mm^2^, which are smaller than those of the device, thus the channel does not affect the measurements with the edge coupler. The flow channel and silicone tubing are joined with a connector, and the solution is injected using a syringe to allow the solution to flow on the device. The flow channel fabrication process is summarized as follows:I.Pouring PDMS into a flow channel mold;II.Removing air bubbles in PDMS with a vacuum pump;III.Heating in an oven at 80 °C for 2 h to harden PDMS;IV.Cutting the flow channel to the size of a single chip;V.Cleaning the chips with ammonia hydrogen peroxide mixture (APM);VI.Wiping the adhesive surface of the flow channel with ethanol-soaked cloth;VII.Ashing the bonding surface of the chip and flow channel;VIII.Attaching the flow channel to the chip;IX.Heating the chip with the flow channel in an oven at 80 °C for 20 min.

## 4. Detection Experiments

### 4.1. Modification with Antibody

Protein detection experiments were performed using the prepared Si MRR. First, the surface of the Si MRR waveguide was modified with antibodies. We used Si-tagged protein G, as shown in Figure 1b. Si-tagged protein G is a fusion protein of the Si tag, which is composed of a peptide that binds strongly to silica particles and is an antibody-binding protein [31]. The use of Si-tag protein G enables the control of the IgG antibody orientation and is expected to improve detection sensitivity [21,32,33].

The IgG modification process using Si-tagged protein G involves the exposure of the Si MRR waveguide to the solutions indicated below.
I.Phosphate-buffered saline (PBS) solution (pH 7.2);II.Si-taged protein G solution (0.1 mg/mL);II.PBS solution;IV.IgG antibody solution (0.01 mg/mL);V.PBS solution.

For the detection of antibodies or target proteins dissolved in pure water or buffer solution, a PDMS flow channel of polydimethylsiloxane was fabricated and set on the Si MRR, as described in Section 3 and shown in Figure 4. A syringe pump is used to maintain a constant flow rate during the experiment. The solution enters from an inlet through a silicone tube, flows over the tip, and exits from the outlet on the other side.

### 4.2. Experiments for Comparison between Specific and Nonspecific Reactions

First, we discuss the bovine serum albumin (BSA) and human serum albumin (HSA) detection experiments conducted to confirm the specific detection using the Si MRR. BSA and HAS adsorb onto the surface of the Si waveguide through IgG antibodies specifically and nonspecifically, respectively. In the case of BSA, the resonant wavelength should shift to the longer wavelength side owing to its specific adsorption onto the surface of the Si waveguide, and in the case of HAS, there should be no wavelength shift because HSA, which is physically adsorbed onto the surface of the waveguide, will be flushed out.

Antibody modification was performed using the fabricated Si MRR and flow channel. PBS was used as a buffer solution. Solution were allowed to flow through the flow channel in the order of protein G solution, PBS, antibody solution, and PBS. PBS is used to flush out unattached proteins.

Figure 5 shows the transmittance spectra of the Si MRR during the antibody modification. The MRR sensor chip was maintained at 20 °C using a Peltier temperature controller during the experiment. After the injection of protein G solution (Step II), a redshift of 0.544 nm was observed. Next, after the injection of PBS, a blueshift of 0.064 nm was measured (Step III). Thirdly, after the injection of IgG antibody solution, a red shift of 0.648 nm was observed (Step IV). Finally, after the injection of PBS, a blueshift of 0.064 nm was observed (Step V). These results show that the resonant wavelength was redshifted by 1.064 nm owing to the antibody modification (Steps I to V).

Next, BSA and HSA were detected. HSA and BSA solutions were alternately injected at increasing concentrations of 1 ng/mL, 10 ng/mL, 100 ng/mL, 1 μg/mL, and 10 μg/mL. The process of measuring each protein concentration was as follows. First, PBS was injected, followed by the protein solution and then PBS. Figure 6 shows the measurement sequence and the resonant wavelengths measured at each protein concentration. For Process a, for example, after the injection of HSA, PBS was injected, and the resonant wavelength for HSA was measured (point X). Next, BSA was injected, followed by PBS injection, and the resonant wavelength was measured for BSA (point Y).

It was found that after the injection of BSA, which shows specific adsorption, the resonant wavelength blueshifted at the concentration of 1 ng/mL (Process a); however, it redshifted at concentrations from 10 ng/mL to 10 μg/mL (Processes b to e). In contrast, the resonant wavelengths after HSA injections were blueshifted at the concentrations from 1 ng/mL to 1 μg/mL (Processes A to D). Redshifts were only observed after BSA was injected, demonstrating that specific detection is possible at any concentration above 10 ng/mL without saturation or physical adsorption. The blueshifts in Processes A, a, B, and C were caused by IgG antibodies desorbed from the Si waveguide surface during injections of the PBS solution, leading to a decrease in the effective refractive index of the Si waveguide. The redshifts in Processes b to e after the injection of BSA were caused by the selective adsorption of BSA onto the surface of the Si waveguide, leading to an increase in the effective refractive index of the Si waveguide.

Figure 7 shows the wavelength shift for each process (Processes A to E and a to e). A wavelength shift is defined as the difference in resonant wavelength measured at the steps before and after the PBS injection. For instance, the wavelength shift of BSA at the concentration of 1 ng/mL (Process a) is the difference in resonant wavelength between points X and Y. The shift of the resonant wavelength is proportional to the change in the effective refractive index of an MRR waveguide on the basis of Equation (1), and the adsorption of antigens onto the Si waveguide surface changes the effective refractive index. In the experimental results shown in Figure 7 the resonance wavelengths redshifted only when BSA was injected, indicating that BSA was selectively detected through the antibody–antigen reaction. Therefore, the selective detection of specific proteins is possible using the Si MRR modified with IgG antibodies.

Although the effective refractive index change should be proportional to the coverage of the antigen on the Si MRR waveguide surface, the experimental results in Figure 7 show that the coverage of BSA adsorbed is not proportional to the concentration of BSA. The relationship between the concentration and coverage of BSA on the Si MRR waveguide requires further study.

To analyze the coverage of the HSA on the surface of the Si waveguide, the theoretical characteristics of the MRR output were calculated using the simulation software Lumerical verFDTD (ver. 8.25), assuming a 20 nm thick antibody layer (15 nm IgG layer + 5 nm Si-tagged protein G layer) [34,35,36] and a 100 nm thick antigen layer for the waveguide shown in Figure 8 In the simulation, a perfectly matched layer (PML) was used as a boundary condition, and the refractive indices of Si and SiO_2_ were assumed to be 3.48 [37] and 1.44 [38], respectively. The refractive indices of antibody and antigen proteins were assumed to be both 1.5 because the refractive index of most proteins is typically in the range of 1.5–1.7 [39,40] and that of influenza A virus is approximately 1.5 [41]. Automatic meshing was used with the minimum mesh step of 0.25 nm. The round-trip length of the MRR and the coupling efficiency were assumed to be 90 μm and 0.05, respectively, which are the same as the design values of the Si MRR used in the experiment. Figure 9 shows the simulation results, which indicate show that a wavelength shift of 1.13 nm is expected when the antibody adheres to the entire waveguide surface. Moreover, a wavelength shift of approximately 0.32 nm is expected when the target attaches to all of the antibodies. On the basis of comparison between the simulated and experimental wavelength shifts, the ratios of the area of the Si waveguide surface to that of the antibody adhering to the waveguide surface and the amount of the antibody adhering to the waveguide surface to that of the antigen that reacted to the antibody were estimated to be approximately 0.92 and 0.49, respectively; namely, the ratio of the area of the waveguide surface to that of the antigen adsorbed onto the waveguide surface was approximately 0.45.

### 4.3. Nucleocapsid Protein Detection Experiment

Nucleocapsid proteins are the shells that encase the genes of coronaviruses and are used as targets in the antigen tests because they are unlikely to change their amino acid sequences owing to mutation or other causes. The nucleocapsid protein detection experiments were performed using the Si MRRs and flow channels. The MRR sensor chip was maintained at 20 °C using a Peltier temperature controller during the experiment.

First, antibody modification was performed, and PBS was used as a buffer solution. Pure water was used to flush out excess protein instead of PBS to reduce the risk of antibodies being flushed out. The sequence of solutions injected for antibody modification was protein A, protein solution, and PBS. Figure 10 shows the transmittance spectra of the Si MRR during antibody modification. The antibody modification process is the same as that discussed in Section 4.1. After the injection of protein A solution, pure water, IgG antibody solution, and PBS, a redshift of 0.313 nm, a blueshift of 0.191 nm, a redshift of 0.689 nm, and a blueshift of 0.219 nm were observed, respectively. Therefore, the redshift of the resonant wavelength due to the antibody modification was 0.592 nm, and the coverage of antibody modification was estimated to be approximately 0.52.

Next, the experiments on nucleocapsid protein solutions were conducted. In the experiments, nucleocapsid protein solutions with concentrations of 100 fg/mL, 1 pg/mL, 10 pg/mL, 100 pg/mL, 1 ng/mL, 10 ng/mL, 100 ng/mL, and 1 μg/mL dissolved in PBS were used. The sequence of solutions injected at each concentration was pure water, nucleocapsid solution, and pure water. Figure 11 shows the resonant wavelengths measured at each nucleocapsid concentration. It was found that although blueshifts of the resonant wavelength at 100 fg/mL and 1 pg/mL were observed (Processes A and B), redshifts were observed from 10 pg/mL to 1 ng/mL in Processes C to E. The redshifts were caused by the adsorption of nucleocapsid proteins onto the surface of the Si waveguide, leading to an increase in the effective refractive index of the Si waveguide. As observed in the BSA detection experiments, the coverage of the nucleocapsid adsorbed onto the surface of the Si MRR waveguide is not proportional to the concentration of the nucleocapsid. The relationship between the concentration of the nucleocapsid protein and its coverage on the Si MRR waveguide requires further study.

Figure 12 shows the transmittance spectra of the Si MRR in Processes C to E. Thereafter, the blueshifts were observed at 10 and 100 ng/mL (Processes F and G). Finally, a redshift was observed at 1 μg/mL (Process H). Figure 13 summarizes the wavelength shifts for each process. The wavelength shift in this experiment is defined as the difference in resonance wavelength measured for pure water before and after the injection of the protein solution. The experimental results show that the nucleocapsid protein was successfully detected without saturation at the concentrations where the redshift of the resonant wavelength was observed in Processes C to E. In particular, the redshift of a resonant wavelength of 27 pm was measured in Process C (nucleocapsid protein concentrations from 1 to 10 pg/mL). Although the experimental results are very preliminary, they show that the potential sensitivity to the nucleocapsid is on the order of 10 pg/mL, which is comparable to that of current antigen tests [42]. The detection time of the Si MRR sensor for each concentration was a few minutes, which is much shorter than that of other antigen tests, such as enzyme-linked immunosorbent assay.

The blueshifts at 100 fg/mL and 1 pg/mL are considered to have occurred because some of the antibodies have been desorbed during Processes A and B, leading to a decrease in the effective refractive index of the Si waveguide. At the point after Process B (after the injection of 1 pg/mL solution), the antibody coverage was estimated to be approximately 0.30 from the blueshift of 0.25 nm during Processes A and B. The cause of the blueshifts at 10 and 100 ng/mL (during Processes F and G) can be explained by the assumption that all antibodies on the MRR waveguide surface were bonded to antigens, and when the solutions with the concentrations of 10 ng/mL and 100 ng/mL were injected, some of the antigens were washed away, and the function as a sensor was restored. This assumption can elucidate the redshift at the concentration of 1 μg/mL.

## 5. Conclusions

We proposed and developed a high-sensitivity silicon MRR optical biosensor for detecting the nucleocapsid protein of SARS-CoV-2. The target protein is selectively detected through the antigen–antibody reaction, which involves a specific adsorption reaction between the antigen and the antibody. The surface of a silicon waveguide is modified with the antibodies, and the target protein is detected by measuring the resonant wavelength shift of an MRR caused by the adsorption of the antigen to the surface of the silicon waveguide. We fabricated silicon MRRs with a round-trip length of 90 μm and a Q-factor of 20,000 on an SOI substrate with PDMS flow channels and characterized their detection characteristics. Firstly, the specific detection of the protein by the MRR sensor is confirmed by the detection of BSA and HSA. Next, the detection of the nucleocapsid protein is performed. The nucleocapsid is one of the proteins that consist coronaviruses and is used as a target in the antigen test. The waveguide surface is modified with the IgG antibodies through Si-tagged protein G. Although the experimental results are very preliminary, they show that the potential sensitivity to nucleocapsids is on the order of 10 pg/mL, which is comparable to the sensitivity of current antigen tests. The experimental results show that the Si MRR biosensors are promising as highly sensitive nucleocapsid protein sensors for detecting SARS-CoV-2.

## Figures and Tables

**Figure 1 sensors-24-03250-f001:**
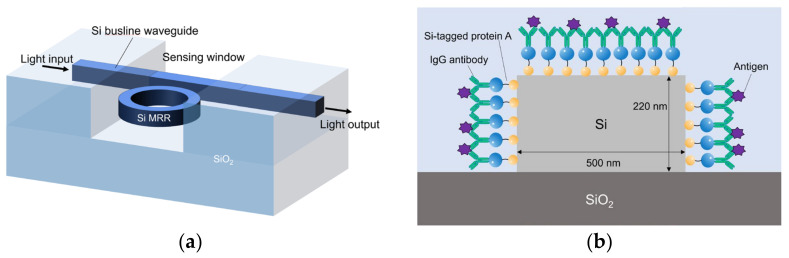
(**a**) Schematic overall view of Si-MRR biosensor. (**b**) Cross-sectional view of Si waveguide with Si-tagged protein G and IgG antibodies.

**Figure 2 sensors-24-03250-f002:**
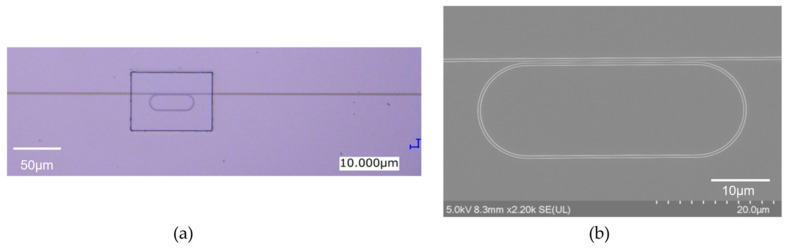
(**a**) Optical microscopy image and (**b**) SEM image of fabricated Si MRR.

**Figure 3 sensors-24-03250-f003:**
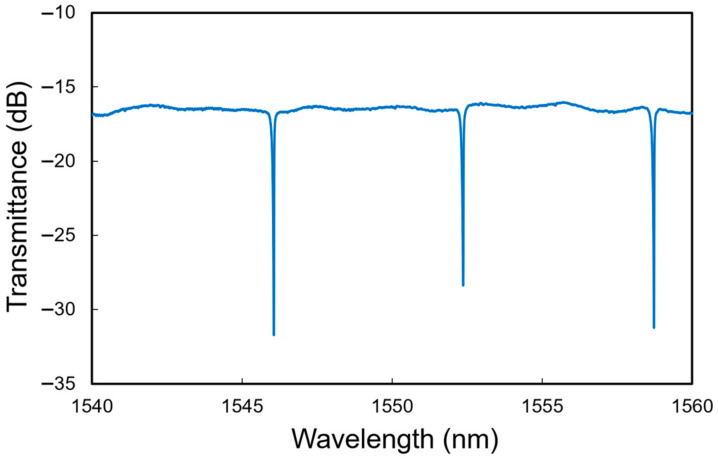
Measured transmittance spectrum of Si MRR.

**Figure 4 sensors-24-03250-f004:**
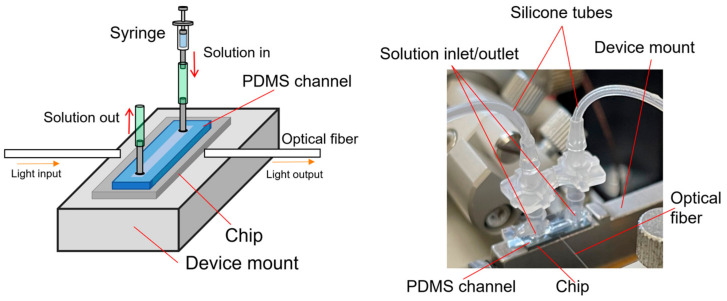
Schematic and photograph of Si MRR sensor chip on device mount for measurement of sensing characteristics.

**Figure 5 sensors-24-03250-f005:**
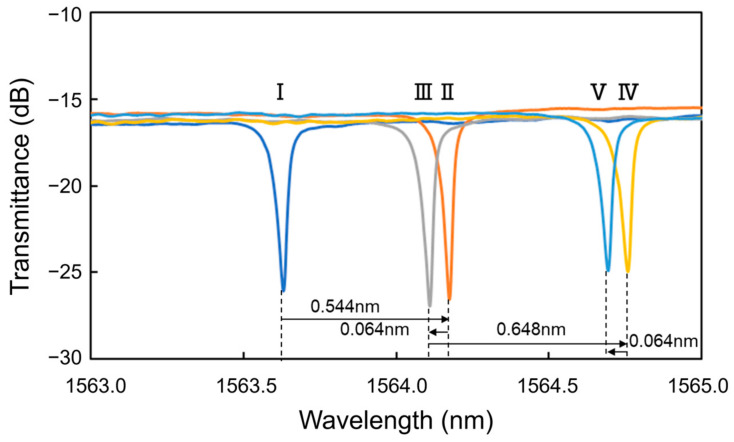
Transmittance spectra of Si MRR during antibody modification.

**Figure 6 sensors-24-03250-f006:**
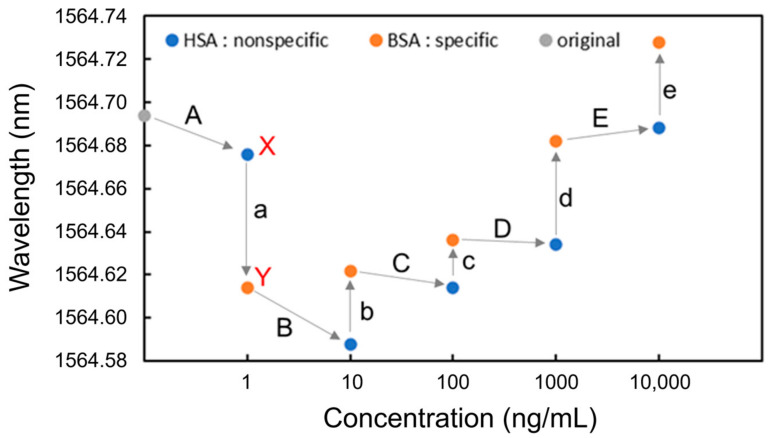
Resonant wavelengths measured at each concentration of BSA and HAS for detection.

**Figure 7 sensors-24-03250-f007:**
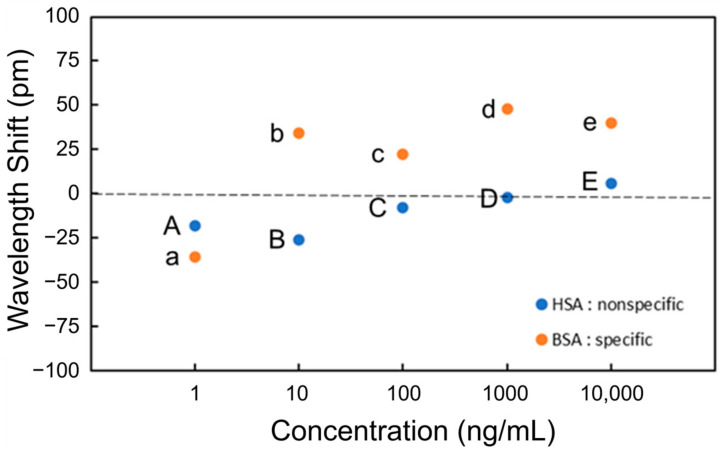
Wavelength shift at each concentration of BSA and HAS for detection.

**Figure 8 sensors-24-03250-f008:**
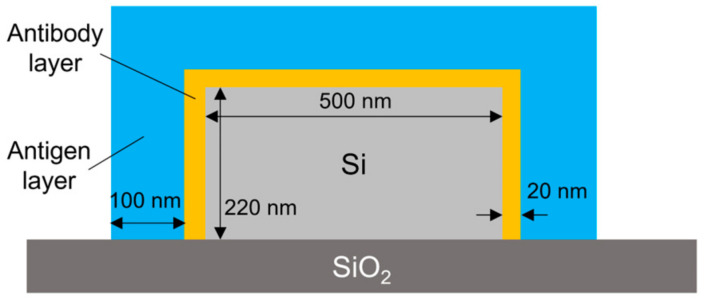
Simulation model.

**Figure 9 sensors-24-03250-f009:**
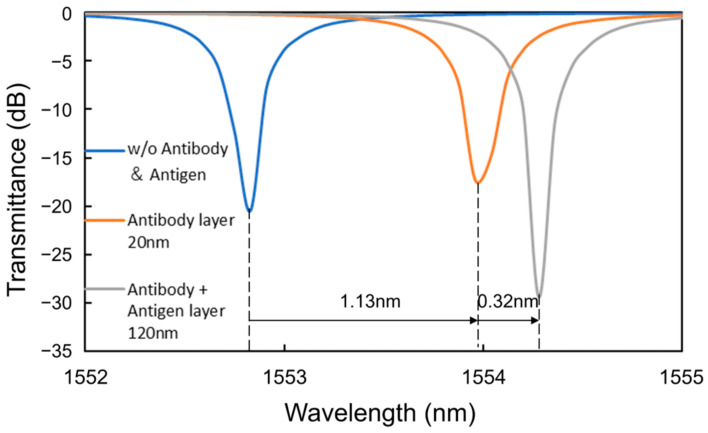
Simulated wavelength shifts caused by antibody and antigen layers.

**Figure 10 sensors-24-03250-f010:**
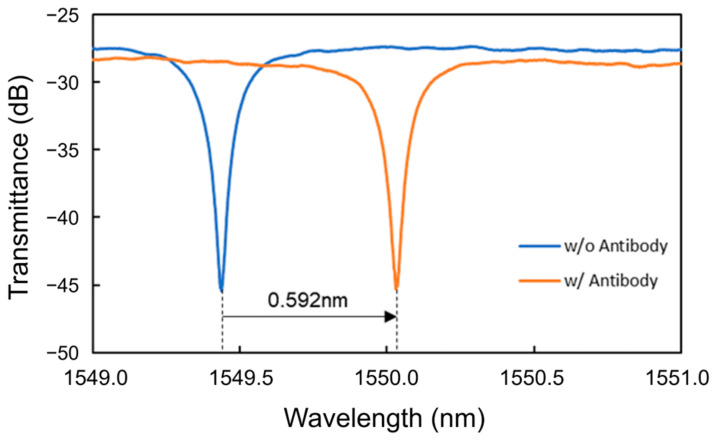
Transmittance spectra of the Si MRR during antibody modification.

**Figure 11 sensors-24-03250-f011:**
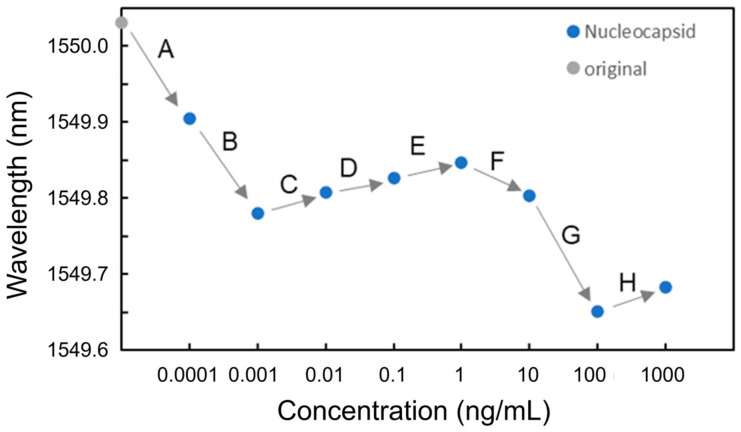
Resonant wavelengths measured at each nucleocapsid concentration.

**Figure 12 sensors-24-03250-f012:**
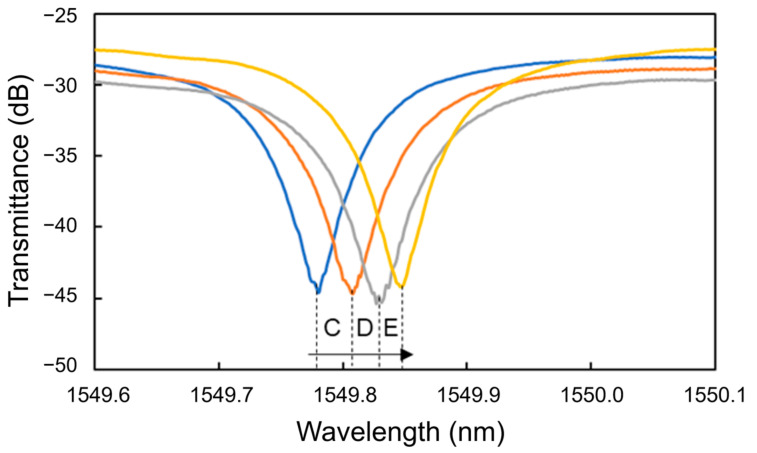
Transmittance spectra of Si MRR during nucleocapsid detection.

**Figure 13 sensors-24-03250-f013:**
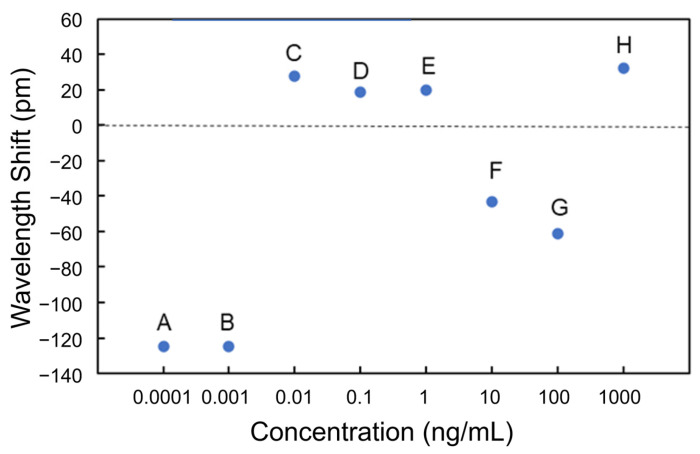
Wavelength shifts measured at each nucleocapsid concentration.

## Data Availability

Data are contained within the article.

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
