# Peer review of "Silicon Microring Resonator Biosensor for Detection of Nucleocapsid Protein of SARS-CoV-2"

_sensors, 2024, doi:10.3390/s24103250_

Round 1

Reviewer 1 Report

Comments and Suggestions for Authors

Comments to author

In this work, the authors present a silicon micro-ring resonator as a biosensor for the detection of Nucleocapsid Protein of SARS‐CoV‐2.

Comment 1:

In line 12, in the abstract section; the author mentions” A high‐sensitivity silicon microring (MRR) optical biosensor for detecting nucleocapsid protein of SARS‐CoV‐2 is proposed and developed” without presenting any value for the sensitivity through the manuscript.

 Comment 2:

The author should highlight the obtained results of the proposed sensor in the abstract section like sensitivity, quality factor, figure of merit, and detection limit.

 Comment 3:

In line 21, in the abstract section; the author mentions that the sensing procedures include IgG antibody implementation to the surface of the sensor which will require preparation for the detection process.

Comment 4:

In line 23, in the abstract section; the author mentions that “As a result,  nucleocapsid protein with a concentration as low as 10 pg/mL is successfully detected, which is comparable to the sensitivity of current antigen tests”. Therefore, what is the novelty of your sensor if it is comparable to other antigen tests like ELISA?

  Comment 5:

The author should present a recent literature review about silicon microring resonator biosensors.

 Comment 6:

Based on the results of this work and the literature, silicon microring resonator-based biosensors have very low resonance wavelength shift in the order of (pm). As a result, a very low sensitivity.

  [1] Michael Dubrovsky, Morgan Blevins, Svetlana V. Boriskina, and Diedrik Vermeulen, "High contrast cleavage detection," Opt. Lett. 46, 2593-2596 (2021), https://doi.org/10.1364/OL.424858.

 [2] Mi Kyoung Park, Jack Sheng Kee, Jessie Yiying Quah, Vivian Netto, Junfeng Song, Qing Fang, Eric Mouchel La Fosse, Guo-Qiang Lo, Label-free aptamer sensor based on silicon microring resonators, Sensors and Actuators B: Chemical, 176,2013, Pp. 552-559, https://doi.org/10.1016/j.snb.2012.08.078.

 Comment 7:

The sensor design has no novelty compared to the published papers based on silicon microring resonator biosensors.

 [2] Mi Kyoung Park, Jack Sheng Kee, Jessie Yiying Quah, Vivian Netto, Junfeng Song, Qing Fang, Eric Mouchel La Fosse, Guo-Qiang Lo, Label-free aptamer sensor based on silicon microring resonators, Sensors and Actuators B: Chemical, 176,2013, Pp. 552-559, https://doi.org/10.1016/j.snb.2012.08.078.

 [3] Caterina Ciminelli, Francesco Dell'Olio, Donato Conteduca, Donato Conteduca, Mario Nicola Armenise, Silicon photonic biosensors, IET Optoelectronics 13(2), (2019), https://doi.org/10.1049/iet-opt.2018.5082.

 [4] Liaquat Ali, Mahrukh Khan, Ma Chaudhry, Liaquat Ali, Mahrukh Khan, +2 authors Ma Chaudhry, IEEE Nanotechnology Symposium (ANTS), (2018), https://doi.org/10.1109/NANOTECH.2018.8653557.

 [5] Zeng, D.; Liu, Q.; Mei, C.; Li, H.; Huang, Q.; Zhang, X. Demonstration of Ultra-High-Q Silicon Microring Resonators for Nonlinear Integrated Photonics. Micromachines 2022, 13, 1155. https://doi.org/10.3390/mi13071155.

 Comment 9:

Since this work is based on a numerical simulation. There are no details about the simulation parameters like boundary conditions, wavelength-dependent refractive index of the materials and the analyte, and meshing/ cell discretization parameters.

 Comment 10:

In Figures 7, and 8, why the resonance wavelengths are red-shifted and blue-shifted as the HAS, and BSA concentration changes?

  According to all these comments, I found that this work can be recommended to be published in the Sensors (MDPI) journal after Major Revision for these comments.

Reviewer 2 Report

Comments and Suggestions for Authors

The present paper consists of an optical Silicon Microring Resonator Biosensor for Detection of Nucleocapsid Protein of SARS‐CoV‐2. This paper presents an optical biosensor with a good sensitivity to SARS‐CoV‐2. The paper is well written and the biosensor is well described, where proper testes were made. The bibliography is not extensive but enough.

Only 2 details to address: The keywords are missing and Figure 2, where the scale and scale units must be checked, special for Figure 2 (a).  

Reviewer 3 Report

Comments and Suggestions for Authors

The article “Silicon Microring Resonator Biosensor for Detection of Nucleocapsid Protein of SARS‐CoV‐2” is dedicated to the developing of the photonic sensor the nucleocapsid protein of SARS‐CoV‐2.

The topic is of special importance and will be interesting for a wide audience. 

While the research provides detailed insights, especially regarding the fabrication process, there are areas that require attention.

1)    Line 84: 210 nm thickness in the text and 220 nm in the Fig. 1

2)    Line 106: MRR does not consist of a ring and waveguide, according to abbreviation MRR was defined in line 60. 

3)    Line 150-151: Therefore, a sensing window is designed so that the cladding becomes air only at the top of the MRR. It is hard to understand. Does it mean that only the top surface is free of cladding? Or microresonator does not surrounded by the cladding as presented in Fig. 1?

4)    Line 181: It is not fully clear what do authors call as flow channel at this point. Fig 5 should be closer to the section.

5)    Fig. 4. Is the same as Fig. 1 (b), consider the possibility mark protein and antigen in Fig. 1 and remove Fig. 4.

6)    Section 4.2 (lines 262-267). The explanation of the results is needed. It is not clear to me that the concentration of only BSA can be detected definitely. Same frequency shifts can lead to the different situations. Also, the mechanisms of the refractive index changes throughout the stages of the experiment should be discussed (considering Eq. 1). That leads to misunderstanding of the results presented in section 4.3.

7) The eigenfrequencies of the microresonator are depend on the temperature of the chip. Do you consider adding thermal stabilisation?

I recommend to clarify the logic of the experiment by highlighting the challenges authors struggling with and how crucial parameters are determined (possible saturation, sensitivity, what may cause the mistakes etc). Introducing this information in the introduction can help set the context for the research and improve overall comprehension.

In summary, the study appears to be sufficiently detailed, particularly in terms of the fabrication process. But the presentation is complicated and hard to understand, the results should be clarified more carefully. I recommend publication of the article after revising the presentation of results to enhance clarity.

Comments on the Quality of English Language

No specific issues found.

Reviewer 4 Report

Comments and Suggestions for Authors

I read with great interest the manuscript titled "Silicon Microring Resonator Biosensor for Detection of Nucleocapsid Protein of SARS‐CoV‐2" by Y. Uchida et al. The subject matter is highly intriguing and aligns well with the scope of topics relevant to "Sensors" and its readership, but it is necessary to raise some important issues.

In the article, the authors describe an optical biosensor based on silicon microring resonators (MRR) for detecting the N protein of SARS-CoV-2. In this manuscript, the authors describe how it was designed and its key parameters. Additionally, they describe the fabrication procedure of the sensor itself and the PDMS fluidics used for the experiments, although it is not very clear how the fluidics are integrated with the MRR, in particular, which is the flow direction on the MRR. Although these parts are appropriately and satisfactorily described, it is not very clear how the measurement setup is structured. What are the main characteristics of the light source? How is it coupled to the fiber? What instrumentation is used to acquire the transmitted light radiation from the fiber? What is the spectral resolution of the instrument?

In the first experimental part, the authors demonstrate how the system is capable of specifically binding the BSA protein rather than HSA. It is my opinion that the results are presented in an approximate and scientifically incorrect manner. It is unclear how many measurements were taken. How reproducible are the experimental results shown in Figure 7? How is the data acquisition performed? Observing Figure 7 (the same applies to Figure 12), it appears that the experimental points are the result of a single acquisition. If not, the authors should provide an assessment of the variability of the spectral resonance position from chip to chip and when measurement conditions (temperature, solution, etc.) are kept constant in the fluidic cell. Essentially, each experimental point should be associated with an error bar. Also referring to Figure 7, the authors should explain the reason for a blue shift when the first solution "A" is injected into the cell.

Moreover, to better understand what happens on the sensor, the authors report some numerical simulations performed in Lumerical, but what puzzles me is the thickness of biological material placed on the sensor's surface. An IgG antibody should not be larger than 10 nm, similarly for BSA (perhaps a bit smaller). The authors should clarify the choice of 100 nm thickness for the biological layers by inserting some bibliographic references.

In the second experimental part, section 4.3, the authors describe experiments carried out with different solutions of N protein, but it is not clear, first, whether the protein is dissolved in water or PBS. Secondly, and more importantly, the interpretation given to the data collected during the experiments is not at all clear. Here too, some questions arise spontaneously. Are the reported results the outcome of a single measurement? What is the measurement error associated with each experimental point? The authors do not provide sufficient explanations regarding why a blue shift is observed first, then a red shift, then again, a blue shift, and then again, a red shift. In the C-E interval, which the authors associate with the detection of the N protein, the observed shifts are in the order of a few pm and, assuming they are beyond the spectral resolution of the spectrophotometer used, they are all of the same magnitudes despite increasing the N protein concentration by 10 times. Finally, the authors claim to have achieved a sensitivity of 10 pg/mL, but I do not believe that the reported measurements are sufficient to support such a claim.

Although promising results are presented in the article, several issues need to be clarified before it can be published in the present journal. In conclusion, I believe the manuscript is suitable for publication in this journal after major revisions.

Reviewer 5 Report

Comments and Suggestions for Authors

This paper proposes a high-sensitivity optical biosensor based on a silicon microring resonator (MRR) for detecting SARS-CoV-2 virus nucleocapsid protein. First, the authors use the characteristics of silicon microring resonators to detect the target protein by measuring the change of resonant wavelength, which has high sensitivity and specificity. As an optical device, silicon microring resonator has a broad application prospect in biosensing. Hence, this paper's research direction is of great scientific significance. Secondly, a silicon microring resonator was fabricated on a silicon insulator (SOI) substrate by a CMOS-compatible manufacturing process, and polydimethylsiloxane (PDMS) runner was used to study the detection characteristics. This process method is repeatable and expandable, providing practical application possibilities. In addition, the authors verified the sensor's selective detection capability through comparative experiments. They successfully detected nucleocapsid proteins with concentrations as low as 10 pg/mL, comparable to the sensitivity of current antigen detection. The sensor has potential practical value and is significant for virus research.

Overall, the work of this paper is interesting. However, there are some problems to be further improved as well:

(1) Some details are not fully described When describing the experimental methods and results. For example, the specific preparation process of the silicon microring resonator, the modification method of the IgG antibody, the concentration and pH of the PBS solution, etc., may cause readers to doubt the repeatability of the experiment.

(2) In terms of keyword selection, it is suggested that the author add some keywords related to silicon microring resonators, optical biosensing, etc., better to reflect the research characteristics and focus of the paper.

(3) Some details need to be worked out. For example, Figure 1 (b) shows the size of the Si‐MRR biosensor at 220X500 nm. The text states, "The thickness and width of the Si waveguide are 210 and 500 nm, respectively." The first letters in Figure 14 should be capitalized.

Comments on the Quality of English Language

Minor editing of English language required.

Round 2

Reviewer 4 Report

Comments and Suggestions for Authors

The latest version of the manuscript shows improvement compared to the previous one, although some parts could still be further enhanced. In my opinion, more investigations and experiments should be provided to improve its overall evaluation. Specifically, I believe the reported measurement is insufficient to claim that the system's limit of detection is 10 pg/mL (in the abstract, the authors wrote 10pg/m, where an "L" is missing). Therefore, I recommend that the authors remove that claim from the abstract and clarify that this is a very preliminary result.

Additional comments:

1) In the introduction section, perhaps the authors should also cite the following manuscript: https://doi.org/10.1016/j.biosx.2023.100413, where a sensor based on Bloch surface waves is used to detect antibodies against the spike protein of SARS-CoV-2. The sensor working basic principles are quite similar to SPR techniques.

2) In lines 78 and 91, the authors wrote "InG"; perhaps they intended to write "IgG".

3) Line 198. The authors report the detection limit for environmental refractive index change "10-4"; in my opinion, they should add the unit RIU to the value 10-4 (10-4 RIU).

4) Line 327. The authors should provide a reference for the refractive index values used in the simulations.

In conclusion, apart from these minor issues, I consider the manuscript eligible for publication in the journal "Sensors".

Author Response

Please find attached the response letter.
